# EIGENGAME: PCA AS A NASH EQUILIBRIUM

**Ian Gemp, Brian McWilliams, Claire Vernade & Thore Graepel**
DeepMind
{imgemp,bmcw,vernade,thore}@google.com

## ABSTRACT

We present a novel view on principal component analysis (PCA) as a competitive game in which each approximate eigenvector is controlled by a player whose goal is to maximize their own utility function. We analyze the properties of this PCA game and the behavior of its gradient based updates. The resulting algorithm—which combines elements from Oja's rule with a generalized Gram-Schmidt orthogonalization—is naturally decentralized and hence parallelizable through message passing. We demonstrate the scalability of the algorithm with experiments on large image datasets and neural network activations. We discuss how this new view of PCA as a differentiable game can lead to further algorithmic developments and insights.

## 1 INTRODUCTION

The *principal components* of data are the vectors that align with the directions of maximum variance. These have two main purposes: a) as interpretable features and b) for data compression. Recent methods for principal component analysis (PCA) focus on the latter, explicitly stating objectives to find the $k$-dimensional subspace that captures maximum variance (e.g., (Tang, 2019)), and leaving the problem of rotating within this subspace to, for example, a more efficient downstream singular value (SVD) decomposition step[1]. This point is subtle, yet critical. For example, any pair of two-dimensional, orthogonal vectors spans all of $\mathbb{R}^2$ and, therefore, captures maximum variance of any two-dimensional dataset. However, for these vectors to be principal components, they must, in addition, align with the directions of maximum variance which depends on the covariance of the data. By learning the optimal subspace, rather than the principal components themselves, objectives focused on subspace error ignore the first purpose of PCA. In contrast, modern nonlinear representation learning techniques focus on learning features that are both disentangled (uncorrelated) and low dimensional (Chen et al., 2016; Mathieu et al., 2018; Locatello et al., 2019; Sarhan et al., 2019).

It is well known that the PCA solution of the $d$-dimensional dataset $X \in \mathbb{R}^{n \times d}$ is given by the eigenvectors of $X^\top X$ or equivalently, the right singular vectors of $X$. Impractically, the cost of computing the full SVD scales with $\mathcal{O}(\min\{nd^2, n^2d\})$-time and $\mathcal{O}(nd)$-space (Shamir, 2015; Tang, 2019). For moderately sized data, randomized methods can be used (Halko et al., 2011). Beyond this, stochastic—or online—methods based on Oja's rule (Oja, 1982) or power iterations (Rutishauser, 1971) are common. Another option is to use *streaming k-PCA* algorithms such as Frequent Directions (FD) (Ghashami et al., 2016) or Oja's algorithm[2] (Allen-Zhu and Li, 2017) with storage complexity $\mathcal{O}(kd)$. Sampling or sketching methods also scale well, but again, focus on the top-$k$ subspace (Sarlos, 2006; Cohen et al., 2017; Feldman et al., 2020).

In contrast to these approaches, we view each principal component (equivalently eigenvector) as a player in a game whose objective is to maximize their own local utility function in controlled competition with other vectors. The proposed utility gradients are interpretable as a combination of Oja's rule and a generalized Gram-Schmidt process. We make the following contributions:

- A novel formulation of PCA as finding the Nash equilibrium of a suitable game,
- A sequential, globally convergent algorithm for approximating the Nash on full-batch data,

---

[1]After learning the top-$k$ subspace $V \in \mathbb{R}^{d \times k}$, the rotation can be recovered via an SVD of $XV$.
[2]FD approximates the top-$k$ subspace; Oja's algorithm approximates the top-$k$ eigenvectors.

- A decentralized algorithm with experiments demonstrating the approach as competitive with modern streaming $k$-PCA algorithms on synthetic and real data,

- In demonstration of the scaling of the approach, we compute the top-32 principal components of the matrix of RESNET-200 activations on the IMAGENET dataset ($n \approx 10^6$, $d \approx 20 \cdot 10^6$).

Each of these contributions is important. Novel formulations often lead to deeper understanding of problems, thereby, opening doors to improved techniques. In particular, $k$-player games are in general complex and hard to analyze. In contrast, PCA has been well-studied. By combining the two fields we hope to develop useful analytical tools. Our specific formulation is important because it obviates the need for any centralized orthonormalization step and lends itself naturally to decentralization. And lastly, theory and experiments support the viability of this approach for continued research.

## 2    PCA AS AN EIGEN-GAME

We adhere to the following notation. Vectors and matrices meant to approximate principal components (equivalently eigenvectors) are designated with hats, $\hat{v}$ and $\hat{V}$ respectively, whereas true principal components are $v$ and $V$. Subscripts indicate which eigenvalue a vector is associated with. For example, $v_i$ is the $i$th largest eigenvector. In this work, we will assume each eigenvalue is distinct. By an abuse of notation, $v_{j<i}$ refers to the set of vectors $\{v_j | j \in \{1, \ldots, i-1\}\}$ and are also referred to as the parents of $v_i$ ($v_i$ is their child). Sums over indices should be clear from context, e.g., $\sum_{j<i} = \sum_{j=1}^{i-1}$. The Euclidean inner product is written $\langle u, v \rangle = u^\top v$. We denote the unit sphere by $\mathcal{S}^{d-1}$ and simplex by $\Delta^{d-1}$ in $d$-dimensional ambient space.

**Outline of derivation**    As argued in the introduction, the PCA problem is often *mis*-interpreted as learning a projection of the data into a subspace that captures maximum variance (equiv. maximizing the trace of a suitable matrix $R$ introduced below). This is in contrast to the original goal of learning the *principal components*. We first develop the intuition for deriving our utility functions by (i) showing that only maximizing the trace of $R$ is not sufficient for recovering **all** principal components (equiv. eigenvectors), and (ii) showing that minimizing off-diagonal terms in $R$ is a complementary objective to maximizing the trace and can recover **all** components. We then consider learning only the top-$k$ and construct utilities that are consistent with findings in (i) and (ii), equal the true eigenvalues at the Nash of the game we construct, and result in a game that is amenable to analysis.

**Derivation of player utilities.**    The *eigenvalue* problem for a symmetric matrix $X^\top X = M \in \mathbb{R}^{d \times d}$ is to find a matrix of $d$ orthonormal column vectors $V$ (implies $V$ is full-rank) such that $MV = V\Lambda$ with $\Lambda$ diagonal. Given a solution to this problem, the columns of $V$ are known as eigenvectors and corresponding entries in $\Lambda$ are eigenvalues. By left-multiplying by $V^\top$ and recalling $V^\top V = VV^\top = I$ by orthonormality (i.e., $V$ is unitary), we can rewrite the equality as

$$V^\top M V = V^\top V \Lambda \stackrel{\text{unitary}}{=} \Lambda. \tag{1}$$

Let $\hat{V}$ denote a guess or estimate of the true eigenvectors $V$ and define $R(\hat{V}) \stackrel{\text{def}}{=} \hat{V}^\top M \hat{V}$. The PCA problem is often posed as maximizing the trace of $R$ (equiv. minimizing reconstruction error):

$$\max_{\hat{V}^\top \hat{V} = I} \left\{ \sum_i R_{ii} = \text{Tr}(R) = \text{Tr}(\hat{V}^\top M \hat{V}) = \text{Tr}(\hat{V}\hat{V}^\top M) = \text{Tr}(M) \right\}. \tag{2}$$

Surprisingly, the objective in (2) is independent of $\hat{V}$, so it cannot be used to recover **all** (i.e., $k = d$) the eigenvectors of $M$—(i). Alternatively, Equation (1) implies the *eigenvalue problem* can be phrased as ensuring all off-diagonal terms of $R$ are zero, thereby ensuring $R$ is diagonal—(ii):

$$\min_{\hat{V}^\top \hat{V} = I} \sum_{i \neq j} R_{ij}^2. \tag{3}$$

It is worth further examining the entries of $R$ in detail. Diagonal entries $R_{ii} = \langle \hat{v}_i, M\hat{v}_i \rangle$ are recognized as *Rayleigh quotients* because $||\hat{v}_i|| = 1$ by the constraints. Off-diagonal entries $R_{ij} = \langle \hat{v}_i, M\hat{v}_j \rangle$ measure alignment between $\hat{v}_i$ and $\hat{v}_j$ under a generalized inner product $\langle \cdot, \cdot \rangle_M$.

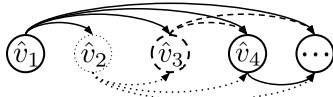

Figure 1: Each player $i$'s utility function depends on its parents represented here by a directed acyclic graph. Each parent must broadcast its vector, "location", down the hierarchy in a fixed order.

So far, we have considered learning all the eigenvectors. If we repeat the logic for the top-$k$ eigenvectors with $k < d$, then by Equation (1), $R$ must still be diagonal. $V$ is not square, so $VV^\top \neq I$, but assuming $V$ is orthonormal as before, we have $VV^\top = P$ is a projection matrix. Left-multiplying Equation (1) by $V$ now reads $(PM)V = V\Lambda$ so we are solving an *eigenvalue problem* for a subspace of $M$.

If we only desire the top-$k$ eigenvectors, maximizing the trace encourages learning a subspace spanned by the top-$k$ eigenvectors, but does not recover the eigenvectors themselves. On the other hand, Equation (3) places no preference on recovering large over small eigenvectors, but does enforce the columns of $\hat{V}$ to actually be eigenvectors. The preceding exercise is intended to introduce minimizing the off-diagonal terms of $R$ as a possible complementary objective for solving top-$k$ PCA. Next, we will use these two objectives to construct utility functions for each eigenvector $\hat{v}_i$.

We want to combine the objectives to take advantage of both their strengths. A valid proposal is

$$\max_{\hat{V}^\top \hat{V}=I} \sum_i R_{ii} - \sum_{i \neq j} R_{ij}^2. \tag{4}$$

However, this objective ignores the natural hierarchy of the top-$k$ eigenvectors. For example, $\hat{v}_1$ is penalized for aligning with $\hat{v}_k$ and vice versa, but $\hat{v}_1$, being the estimate of the largest eigenvector, should be free to search for the direction that captures the most variance independent of the locations of the other vectors. Instead, first consider solving for the top-1 eigenvector, $v_1$, in which case $R = [\langle \hat{v}_1, M\hat{v}_1 \rangle]$ is a $1 \times 1$ matrix. In this setting, Equation (3) is not applicable because there are no off-diagonal elements, so $\max_{\hat{v}_1^\top \hat{v}_1 = 1} \langle \hat{v}_1, M\hat{v}_1 \rangle$ is a sensible utility function for $\hat{v}_1$.

If considering the top-2 eigenvectors, $\hat{v}_1$'s utility remains as before, and we introduce a new utility for $\hat{v}_2$. Equation (3) is now applicable, so $\hat{v}_2$'s utility is

$$\max_{\hat{v}_2^\top \hat{v}_2 = 1, \hat{v}_1^\top \hat{v}_2 = 0} \langle \hat{v}_2, M\hat{v}_2 \rangle - \frac{\langle \hat{v}_2, M\hat{v}_1 \rangle^2}{\langle \hat{v}_1, M\hat{v}_1 \rangle} \tag{5}$$

where we have divided the off-diagonal penalty by $\langle v_1, Mv_1 \rangle$ so a) the two terms in Equation (5) are on a similar scale and b) for reasons that ease analysis. Additionally note that the constraint $\hat{v}_1^\top \hat{v}_2 = 0$ may be redundant at the optimum ($\hat{v}_1^* = v_1, \hat{v}_2^* = v_2$) because the second term, $\langle \hat{v}_2^*, M\hat{v}_1^* \rangle^2 = \langle v_2, Mv_1 \rangle^2 = \Lambda_{11}^2 \langle v_2, v_1 \rangle^2$, already penalizes such deviations ($\Lambda_{ii}$ is the $i$th largest eigenvector). These reasons motivate the following set of objectives (utilities), one for each vector $i \in \{1, \ldots, k\}$:

$$\max_{\hat{v}_i^\top \hat{v}_i = 1} \left\{ u_i(\hat{v}_i | \hat{v}_{j<i}) = \hat{v}_i^\top M\hat{v}_i - \sum_{j<i} \frac{(\hat{v}_i^\top M\hat{v}_j)^2}{\hat{v}_j^\top M\hat{v}_j} = ||X\hat{v}_i||^2 - \sum_{j<i} \frac{\langle X\hat{v}_i, X\hat{v}_j \rangle^2}{\langle X\hat{v}_j, X\hat{v}_j \rangle} \right\} \tag{6}$$

where the notation $u_i(a_i | b)$ emphasizes that player $i$ adjusts $a_i$ to maximize a utility conditioned on $b$.

It is interesting to note that by incorporating knowledge of the natural hierarchy (see Figure 1), we are immediately led to constructing asymmetric utilities, and thereby, inspired to formulate the PCA problem as a game, rather than a direct optimization problem as in Equation (4).

A key concept in games is a Nash equilibrium. A Nash equilibrium specifies a variable for each player from which no player can unilaterally deviate and improve their outcome. In this case, $\hat{V}$ is a (strict-)Nash equilibrium if and only if for all $i$, $u_i(\hat{v}_i | \hat{v}_{j<i}) > u_i(z_i | \hat{v}_{j<i})$ for all $z_i \in \mathcal{S}^{d-1}$.

**Theorem 2.1** (**PCA Solution is the Unique strict-Nash Equilibrium**). *Assume that the top-$k$ eigenvalues of $X^\top X$ are positive and distinct. Then the top-$k$ eigenvectors form the unique strict-Nash equilibrium of the proposed game in Equation (6).*[3] *The proof is deferred to Appendix L.*

Solving for the Nash of a game is difficult in general. Specifically, it belongs to the class of PPAD-complete problems (Gilboa and Zemel, 1989; Daskalakis et al., 2009). However, because the game

is hierarchical and each player's utility only depends on its parents, it is possible to construct a sequential algorithm that is convergent by solving each player's optimization problem in sequence.

## 3   METHOD

**Utility gradient.**   In Section 2, we mentioned that normalizing the penalty term from Equation (5) had a motivation beyond scaling. Dividing by $\langle \hat{v}_j, M\hat{v}_j \rangle$ results in the following gradient for player $i$:

$$\nabla_{\hat{v}_i} u_i(\hat{v}_i | \hat{v}_{j<i}) = 2M \underbrace{\left[ \hat{v}_i - \sum_{j<i} \frac{\hat{v}_i^\top M \hat{v}_j}{\hat{v}_j^\top M \hat{v}_j} \hat{v}_j \right]}_{\text{generalized Gram-Schmidt}} = 2X^\top \left[ X\hat{v}_i - \sum_{j<i} \frac{\langle X\hat{v}_i, X\hat{v}_j \rangle}{\langle X\hat{v}_j, X\hat{v}_j \rangle} X\hat{v}_j \right]. \quad (7)$$

The resulting gradient with normalized penalty term has an intuitive meaning. It consists of a single generalized Gram-Schmidt step followed by the standard matrix product found in power iteration and Oja's rule. Also, notice that applying the gradient as a fixed point operator in sequence ($\hat{v}_i \leftarrow \frac{1}{2} \nabla_{\hat{v}_i} u_i(\hat{v}_i | \hat{v}_{j<i})$) on $M = I$ recovers the standard Gram-Schmidt procedure for orthogonalization.

**A sequential algorithm.**   Each eigenvector can be learned by maximizing its utility. The vectors are constrained to the unit sphere, a non-convex *Riemannian* manifold, so we use *Riemmanian* gradient ascent with gradients given by Equation (7). In this case, Riemannian optimization theory simply requires an intermediate step where the gradient, $\nabla_{\hat{v}_i}$, is projected onto the tangent space of the sphere to compute the Riemannian gradient, $\nabla_{\hat{v}_i}^R$. A more detailed illustration can be found in Appendix J. Recall that each $u_i$ depends on $\hat{v}_{j<i}$. If any of $\hat{v}_{j<i}$ are being learned concurrently, then $\hat{v}_i$ is maximizing a non-stationary objective which makes a convergence proof difficult. Instead, for completeness, we prove convergence assuming each $\hat{v}_i$ is learned in sequence. Algorithm 1 learns $\hat{v}_i$ given fixed parents $\hat{v}_{j<i}$; we present the convergence guarantee in Section 4 and details on setting $\rho_i$ and $\alpha$ in Appendix O.

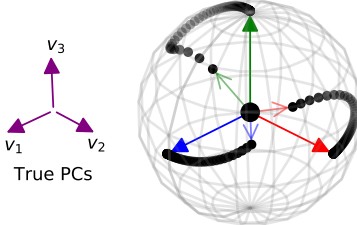

Figure 2: EigenGame guides each $\hat{v}_i$ along the unit-sphere from ↑ to ↟ in parallel; $M = \texttt{diag}([3, 2, 1])$.

---

**Algorithm 1** EigenGame$^R$-Sequential

Given: matrix $X \in \mathbb{R}^{n \times d}$, maximum error tolerance $\rho_i$, initial vector $\hat{v}_i^0 \in \mathcal{S}^{d-1}$, learned approximate parents $\hat{v}_{j<i}$, and step size $\alpha$.
$\hat{v}_i \leftarrow \hat{v}_i^0$
$t_i = \lceil \frac{5}{4} \min(||\nabla_{\hat{v}_i^0} u_i||/2, \rho_i)^{-2} \rceil$
**for** $t = 1 : t_i$ **do**
   $\texttt{rewards} \leftarrow X\hat{v}_i$
   $\texttt{penalties} \leftarrow \sum_{j<i} \frac{\langle X\hat{v}_i, X\hat{v}_j \rangle}{\langle X\hat{v}_j, X\hat{v}_j \rangle} X\hat{v}_j$
   $\nabla_{\hat{v}_i} \leftarrow 2X^\top \left[ \texttt{rewards} - \texttt{penalties} \right]$
   $\nabla_{\hat{v}_i}^R \leftarrow \nabla_{\hat{v}_i} - \langle \nabla_{\hat{v}_i}, \hat{v}_i \rangle \hat{v}_i$
   $\hat{v}_i' \leftarrow \hat{v}_i + \alpha \nabla_{\hat{v}_i}^R$
   $\hat{v}_i \leftarrow \frac{\hat{v}_i'}{||\hat{v}_i'||}$
**end for**
return $\hat{v}_i$

**Algorithm 2** EigenGame$^R$ (EigenGame—update with $\nabla_{\hat{v}_i}$ instead of $\nabla_{\hat{v}_i}^R$)

Given: stream, $X_t \in \mathbb{R}^{m \times d}$, total iterations $T$, initial vector $\hat{v}_i^0 \in \mathcal{S}^{d-1}$, and step size $\alpha$.
$\hat{v}_i \leftarrow \hat{v}_i^0$
**for** $t = 1 : T$ **do**
   $\texttt{rewards} \leftarrow X_t\hat{v}_i$
   $\texttt{penalties} \leftarrow \sum_{j<i} \frac{\langle X_t\hat{v}_i, X_t\hat{v}_j \rangle}{\langle X_t\hat{v}_j, X_t\hat{v}_j \rangle} X_t\hat{v}_j$
   $\nabla_{\hat{v}_i} \leftarrow 2X_t^\top \left[ \texttt{rewards} - \texttt{penalties} \right]$
   $\nabla_{\hat{v}_i}^R \leftarrow \nabla_{\hat{v}_i} - \langle \nabla_{\hat{v}_i}, \hat{v}_i \rangle \hat{v}_i$
   $\hat{v}_i' \leftarrow \hat{v}_i + \alpha \nabla_{\hat{v}_i}^R$
   $\hat{v}_i \leftarrow \frac{\hat{v}_i'}{||\hat{v}_i'||}$
   $\texttt{broadcast}(\hat{v}_i)$
**end for**
return $\hat{v}_i$

---

**A decentralized algorithm.**   While Algorithm 1 enjoys a convergence guarantee, learning every parent $\hat{v}_{j<i}$ before learning $\hat{v}_i$ may be unnecessarily restrictive. Intuitively, as parents approach their respective optima, they become quasi-stationary, so we do not expect maximizing utilities in parallel to be problematic in practice. To this end, we propose Algorithm 2 visualized in Figure 2.

---

[3]Unique up to a sign change; this is expected as both $v_i$ and $-v_i$ represent the same principal component.

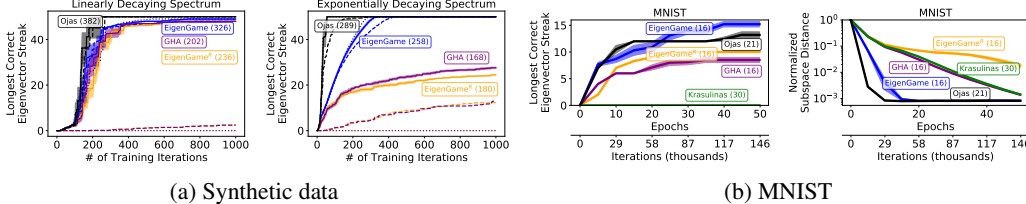

(a) Synthetic data        (b) MNIST

Figure 3: (a) The longest streak of consecutive vectors with angular error less than $\frac{\pi}{8}$ radians is plotted versus algorithm iterations for a matrix $M \in \mathbb{R}^{50 \times 50}$ with a spectrum decaying from 1000 to 1 linearly and exponentially. Average runtimes are reported in milliseconds next to the method names[5]. We omit Krasulina's as it is only designed to find the top-$k$ subspace. Both EigenGame variants and GHA achieve similar asymptotes on the linear spectrum. (b) Longest streak and subspace distance on MNIST with average runtimes reported in seconds. (a,b) Learning rates were chosen from $\{10^{-3}, \dots, 10^{-6}\}$ on 10 held out runs. Solid lines denote results with the best performing learning rate. Dotted and dashed lines denote results using the best learning rate $\times$ 10 and 0.1. All plots show means over 10 trials. Shading highlights $\pm$ standard error of the mean for the best learning rates.

In practice we can assign each eigenvector update to its own device (e.g. a GPU or TPU). Systems with fast interconnects may facilitate tens, hundreds or thousands of accelerators to be used. In such settings, the overhead of broadcast($\hat{v}_i$) is minimal. We can also specify that the data stream is co-located with the update so $\hat{v}_i$ updates with respect to its own $X_{i,t}$. This is a standard paradigm for e.g. data-parallel distributed neural network training. We provide further details in Section 6.

**Message Passing on a DAG.** Our proposed utilities enforce a strict hierarchy on the eigenvectors. This is a simplification that both eases analysis (see Appendix M) and improves convergence[4], however, it is not optimal. We assume vectors are initialized randomly on the sphere and, for instance, $\hat{v}_k$ may be initialized closer to $v_1$ than even $\hat{v}_1$ and vice versa. The hierarchy shown in Figure 1 enforces a strict graph structure for broadcasting information of parents to the childrens' utilities.

To our knowledge, our utility formulation in Equation (6) is novel. One disadvantage is that stochastic gradients of Equation (7) are biased. This is mitigated with large batch sizes (further discussion in Appendix I).

## 4   CONVERGENCE OF EIGENGAME

Here, we first show that Equation (6) has a simple form such that any local maximum of $u_i$ is also a global maximum. Player $i$'s utility depends on its parents, so we next explain how error in the parents propagates to children through mis-specification of player $i$'s utility. Using the first result and accounting for this error, we are then able to give global, finite-sample convergence guarantees in the full-batch setting by leveraging recent non-convex Riemannian optimization theory.

**The utility landscape and parent-to-child error propagation.** Equation (6) is abstruse, but we prove that the shape of player $i$'s utility is simply sinusoidal in the angular deviation of $\hat{v}_i$ from the optimum. The amplitude of the sinusoid varies with the direction of the angular deviation along the unit-sphere and is dependent on the accuracy of players $j < i$. In the special case where players $j < i$ have learned the top-$(i-1)$ eigenvectors exactly, player $i$'s utility simplifies (see Lemma N.1) to

$$u_i(\hat{v}_i, \{v_{j<i}\}) = \Lambda_{ii} - \sin^2(\theta_i)\Big(\Lambda_{ii} - \sum_{l>i} z_l \Lambda_{ll}\Big) \tag{8}$$

where $\theta_i$ is the angular deviation and $z \in \Delta^{d-1}$ parameterizes the deviation direction. Note that $\sin^2$ has period $\pi$ instead of $2\pi$, which simply reflects the fact that $v_i$ and $-v_i$ are both eigenvectors.

---

[4]EigenGame sans order learns max 1 PC and sans order+normalization 5 PCs on data in Figure 3a.

[5]EigenGame runtimes are longer than those of EigenGame$^R$ in the synthetic experiments despite strictly requiring fewer FLOPS; apparently this is due to low-level floating point arithmetic specific to the experiments.

An error propagation analysis reveals that it is critical to learn the parents to a given degree of accuracy. The angular distance between $v_i$ and the maximizer of player $i$'s utility with approximate parents has $\tan^{-1}$ dependence (i.e., a soft step-function; see Lemma N.5 and Figure 13 in Appendix N).

**Theorem 4.1** (**Global convergence**). *Algorithm 1 achieves finite sample convergence to within $\theta_{tol}$ angular error of the top-$k$ principal components, **independent of initialization**. Furthermore, if each $\hat{v}_i$ is initialized to within $\frac{\pi}{4}$ of $v_i$, Algorithm 1 returns the components with angular error less than $\theta_{tol}$*

$$\text{in } T = \left\lceil \mathcal{O}\Big(k\Big[\frac{(k-1)!}{\theta_{tol}} \prod_{i=1}^{k} \Big(\frac{16\Lambda_{11}}{g_i}\Big)\Big]^2\Big)\right\rceil \text{ iterations.}$$ Proofs are deferred to Appendices O.4 and O.5.

Angular error is defined as the angle between $\hat{v}_i$ and $v_i$: $\theta_i = \sin^{-1}(\sqrt{1 - \langle v_i, \hat{v}_i \rangle^2})$. The first $k$ in the formula for $T$ appears from a naive summing of worst case bounds on the number of iterations required to learn each $\hat{v}_{j<k}$ individually. The constant 16 arises from the error propagation analysis; parent vectors, $\hat{v}_{j<i}$, must be learned to under 1/16th of a canonical error threshold, $\frac{g_i}{(i-1)\Lambda_{11}}$, for the child $\hat{v}_i$ where $g_i = \Lambda_{ii} - \Lambda_{i+1,i+1}$. The Riemannian optimization theory we leverage dictates that $\frac{1}{\rho^2}$ iterations are required to meet a $\mathcal{O}(\rho)$ error threshold. This is why the squared inverse of the error threshold appears here. Breaking down the error threshold itself, the ratio $\Lambda_{11}/g_i$ says that more iterations are required to distinguish eigenvectors when the difference between them (summarized by the gap $g_i$) is small relative to the scale of the spectrum, $\Lambda_{11}$. The $(k-1)!$ term appears because learning smaller eigenvectors requires learning a much more accurate $\hat{v}_1$ higher up the DAG.

Lastly, the utility function for each $\hat{v}_i$ is sinusoidal, and it is possible that we initialize $\hat{v}_i$ with initial utility arbitrarily close to the trough (bottom) of the function where gradients are arbitrarily small. This is why the global convergence rate depends on the initialization in general. Note that Algorithm 1 effectively detects the trough by measuring the norm of the initial gradient ($\nabla_{\hat{v}_i^0} u_i$) and scales the number of required iterations appropriately. A complete theorem that considers the probability of initializing $\hat{v}_i$ within $\frac{\pi}{4}$ of $v_i$ is in Appendix O, but this possibility shrinks to zero in high dimensions.

We would also like to highlight that these theoretical findings are strong relative to some other claims. For example, the exponential convergence guarantee for Matrix Krasulina requires the initial guess at the eigenvectors capture the top-$(k-1)$ subspace (Tang, 2019), unlikely when $d \gg k$. A similar condition is required in (Shamir, 2016b). These guarantees are given for the mini-batch setting while ours is for the full-batch, however, we provide global convergence without restrictions on initialization.

## 5 RELATED WORK

PCA is a century-old problem and a massive literature exists (Jolliffe, 2002; Golub and Van Loan, 2012). The standard solution to this problem is to compute the SVD, possibly combined with randomized algorithms, to recover the top-$k$ components as in (Halko et al., 2011) or with Frequent Directions (Ghashami et al., 2016) which combines sketching with SVD.

In neuroscience, Hebb's rule (Hebb, 2005) refers to a connectionist rule that solves for the top eigenvector of a matrix $M$ using additive updates of a vector $v$ as $v \leftarrow v + \eta M v$. Likewise, Oja's rule (Oja, 1982; Shamir, 2015) refers to a similar update $v \leftarrow v + \eta(I - vv^\top)Mv$. In machine learning, using a normalization step of $v \leftarrow v/||v||$ with Hebb's rule is somewhat confusingly referred to as Oja's algorithm (Shamir, 2015), the reason being that the subtractive term in Oja's rule can be viewed as a regularization term for implicitly enforcing the normalization. In the limit of infinite step size, $\eta \to \infty$, Oja's algorithm effectively becomes the well known Power method. If a normalization step is added to Oja's rule, this is referred to as Krasulina's algorithm (Krasulina, 1969). In the language of Riemannian manifolds, $v/||v||$ can be recognized as a retraction and $(I - vv^\top)$ as projecting the gradient $Mv$ onto the tangent space of the sphere (Absil et al., 2009).

Many of the methods above have been generalized to the top-$k$ components. Most generalizations involve adding an orthonormalization step after each update, typically accomplished with a `QR` factorization plus some minor sign accounting (e.g., see Algorithm 3 in Appendix A.1). An extension of Krasulina's algorithm to the top-$k$ setting, termed Matrix Krasulina (Tang, 2019), was recently proposed in the machine learning literature. This algorithm can be recognized as projecting the gradient onto the Stiefel manifold (the space of orthonormal matrices) followed by a `QR` step to maintain orthonormality, which is a well known retraction.

Maintaining orthonormality via QR is computationally expensive. Amid and Warmuth (2019) propose an alternative Krasulina method which does not require re-orthonormalization but instead requires inverting a $k \times k$ matrix; in a streaming setting restricted to minibatches of size 1 ($X_t \in \mathbb{R}^d$), Sherman-Morrison (Golub and Van Loan, 2012) can be used to efficiently replace the inversion step. Raja and Bajwa (2020) develop a data-parallel distributed algorithm for the top eigenvector. Alternatively, the Jacobi eigenvalue algorithm explicitly represents the matrix of eigenvectors as a Givens rotation matrix using sin's and cos's and rotates $M$ until it is diagonal (Golub and Van der Vorst, 2000).

In contrast, other methods extract the top components in sequence by solving for the $i$th component using an algorithm such as power iteration or Oja's, and then enforcing orthogonality by removing the learned subspace from the matrix, a process known as *deflation*. Alternatively, the deflation process may be intertwined with the learning of the top components. The generalized Hebbian algorithm (Sanger, 1989) (GHA) works this way as do Lagrangian inspired formulations (Ghojogh et al., 2019) as well as our own approach. We make the connection between GHA and our algorithm concrete in Prop. K.1. Note, however, that the GHA update is not the gradient of any utility (Prop. K.2) and therefore, lacks a clear game interpretation.

Of these, Oja's algorithm has arguably been the most extensively studied (Shamir, 2016a; Allen-Zhu and Li, 2017)[6] Note that Oja's algorithm converges to the actual principal components (Allen-Zhu and Li, 2017) and Matrix Krasulina (Tang, 2019) converges to the top-$k$ subspace. However, neither can be obviously decentralized. GHA (Sanger, 1989) converges to the principal components asymptotically and can be decentralized (Gang et al., 2019). Each of these is applicable in the streaming $k$-PCA setting.

## 6 EXPERIMENTS

We compare our approach against GHA, Matrix Krasulina, and Oja's algorithm[7]. We present both EigenGame and EigenGame$^R$ which projects the gradient onto the tangent space of the sphere each step. We measure performance of methods in terms of principal component accuracy and subspace distance. We measure principal component accuracy by the number of consecutive components, or *longest streak*, that are estimated within an angle of $\frac{\pi}{8}$ from ground truth. For example, if the angular errors of the $\hat{v}_i$'s returned by a method are, in order, $[\theta_1, \theta_2, \theta_3, \ldots] = [\frac{\pi}{16}, \frac{\pi}{4}, \frac{\pi}{10}, \ldots]$, then the method is credited with a streak of only 1 regardless of the errors $\theta_{i>2}$. For Matrix Krasulina, we first compute the optimal matching from $\hat{v}_i$ to ground truth before measuring angular error. We present the longest streak as opposed to "# of eigenvectors found" because, in practice, no ground truth is available and we think the user should be able to place higher confidence in the larger eigenvectors being correct. If an algorithm returns $k$ vectors, $\frac{k}{2}$ of which are accurate components but does not indicate which, this is less helpful. We measure normalized subspace distance using $1 - \frac{1}{k} \cdot \text{Tr}(U^* P) \in [0, 1]$ where $U^* = V V^\dagger$ and $P = \hat{V} \hat{V}^\dagger$ similarly to Tang (2019).

**Synthetic data.** Experiments on synthetic data demonstrate the viability of our approach (Figure 3a). Oja's algorithm performs best on synthetic experiments because strictly enforcing orthogonalization with an expensive QR step greatly helps when solving for **all** eigenvectors. EigenGame is able to effectively parallelize this over $k$ machines and the advantage of QR diminishes in Figure 3b. The remaining algorithms perform similarly on a linearly decaying spectrum, however, EigenGame performs better on an exponentially decaying spectrum due possibly to instability of Riemannian gradients near the equilibrium (see Appendix J for further discussion). GHA and EigenGame$^R$ are equivalent under specific conditions (see Proposition K.1).

Figure 4a shows EigenGame solves for the eigenvectors up to a high degree of accuracy $\frac{\pi}{32}$, i.e. the convergence results in Figure 3a are not the result of using a *loose* tolerance of $\frac{\pi}{8}$. With the lower tolerance, all algorithms take slightly more iterations to learn the eigenvectors of the linear spectrum; it is difficult to see any performance change for the exponential spectrum. Although Theorem 4.1 assumes distinct eigenvalues, Figure 4b supports the claim that EigenGame does not require distinct eigenvalues for convergence. We leave proving convergence in this setting to future work.

---

[6]See Table 1 in (Allen-Zhu and Li, 2017).

[7]A detailed discussion of Frequent Directions (Ghashami et al., 2016) can be found in Appendix H.

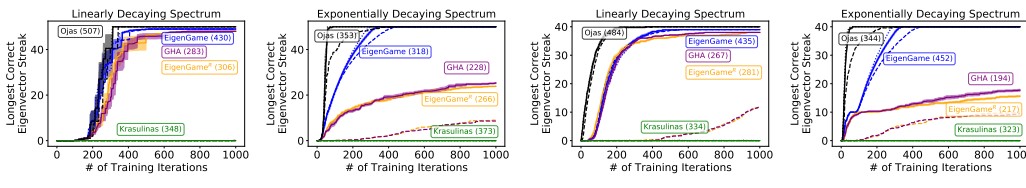

(a) Synthetic data: Stricter Tolerance

(b) Synthetic: Repeated Eigenvalues

Figure 4: (a) Repeats analysis of Figure 3a but for a lower angular tolerance of $\frac{\pi}{32}$. (b) Repeats analysis of Figure 3a with an angular tolerance of $\frac{\pi}{8}$ as before, but with eigenvalues $10 - 19$ of the ordered spectrum overwritten with $\lambda_{10}$ of the original spectrum. We compute angular error for the eigenvectors on either side of this "bubble" to show that EigenGame finds these eigenvectors despite repeated eigenvalues in the spectrum; note $40/50$ is optimal in this experiment.

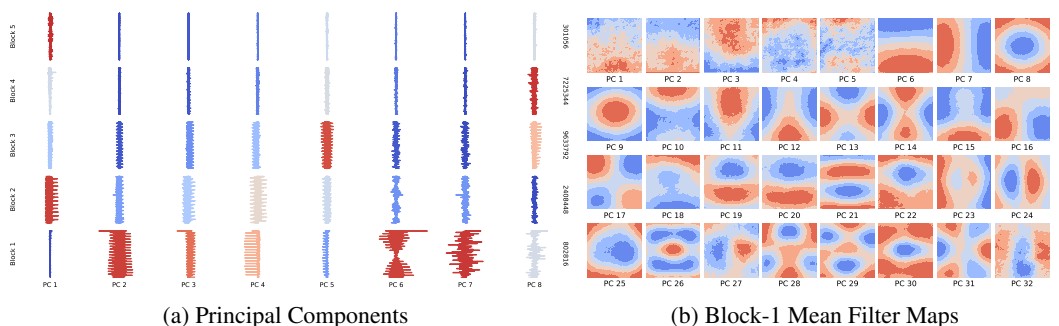

(a) Principal Components

(b) Block-1 Mean Filter Maps

Figure 5: (a) Top-8 principal components of the activations of a RESNET-200 on IMAGENET ordered block-wise by network topology (dimension of each block on the right $y$-axis). Block 1 is closest to input and Block 5 is the output of the network. Color coding is based on relative variance between blocks across the top-8 PCs from blue (low) to red (high). (b) Block 1 mean activation maps of the top-32 principal components of RESNET-200 on IMAGENET computed with EigenGame.

**MNIST handwritten digits.** We compare EigenGame against GHA, Matrix Krasulina, and Oja's algorithm on the MNIST dataset (Figure 3b). We flatten each image in the training set to obtain a $60,000 \times 784$ dimensional matrix. EigenGame is competitive with Oja's in a high batch size regime (1024 samples per mini-batch). The performance gap between EigenGame and the other methods shrinks as the mini-batch size is reduced (see Appendix I), expectedly due to biased gradients.

**The principal components of RESNET-200 activations on IMAGENET are edge filters.** A primary goal of PCA is to obtain *interpretable* low-dimensional representations. To this end we present an example of using EigenGame to compute the top-32 principal components of the activations of a pretrained RESNET-200 on the IMAGENET dataset. We concatenate the flattened activations from the output of each residual block resulting in a $d \approx 20M$ dimensional vector representation for each of the roughly 1.2M input images. It is not possible to store the entire 195TB matrix in memory, nor incrementally compute the Gram/covariance matrix.

We implemented a data-and-model parallel version of EigenGame in JAX (Bradbury et al., 2018) where each $\hat{v}_i$ is assigned to it's own TPU (Jouppi et al., 2017). Each device keeps a local copy of the RESNET parameters and the IMAGENET datastream. Sampling a mini-batch (of size 128), computing the network activations and updating $\hat{v}_i$ are all performed locally. The `broadcast` ($\hat{v}_i$) step is handled by the `pmap` and `lax.all_gather` functions. Computing the top-32 principal components takes approximately nine hours on 32 TPUv3s.

Figure 5a shows the top principal components of the activations of the trained network organized by network topology (consisting of five residual blocks). Note that EigenGame is *not* applied block-wise, but on all 20M dimensions. We do not assume independence between blocks and the eigenvector has unit norm across all blocks. We observe that Block 1 (closest to input) of PC 1 has very small magnitude activations relative to the other PCs. This is because PC 1 should capture the variance which discriminates most between the classes in the dataset. Since Block 1 is mainly

concerned with learning low-level image filters, it stands to reason that although these are important for good performance, they do not necessarily extract abstract representations which are useful for classification. Conversely, we see that PC 1 has larger relative activations in the later blocks.

We visualize the average principal activation in Block $1^8$ in Figure 5b. The higher PCs learn distinct filters (Gabor filters, Laplacian-of-Gaussian filters c.f. (Bell and Sejnowski, 1997)).

## 7 CONCLUSION

*It seems easier to train a bi-directional LSTM with attention than to compute the SVD of a large matrix. –Chris Re*

NeurIPS 2017 Test-of-Time Award, Rahimi and Recht (Rahimi and Recht, 2017).

In this work we motivated PCA from the perspective of a multi-player game. This inspired a decentralized algorithm which enables large-scale principal components estimation. To demonstrate this we used EigenGame to analyze a large neural network through the lens of PCA. To our knowledge this is the first academic analysis of its type and scale (for reference, (Tang, 2019) compute the top-6 PCs of the $d = 2300$ outputs of VGG). EigenGame also opens a variety of research directions.

**Scale.** In experiments, we broadcast across all edges in Figure 1 every iteration. Introducing lag or broadcasting with dropout may improve efficiency. Can we further reduce our memory footprint by storing only scalars of the losses and avoiding congestion through online bandit or reinforcement learning techniques? Our decentralized algorithm may have implications for federated and privacy preserving learning as well (Heinze et al., 2016; Heinze-Deml et al., 2018; Bonawitz et al., 2019).

**Games.** EigenGame has a unique Nash equilibrium due to the fixed DAG structure, but vectors are initialized randomly so $\hat{v}_k$ may start closer to $v_1$ than $\hat{v}_1$ does. Adapting the DAG could make sense, but might also introduce spurious fixed points or suboptimal Nash. Might replacing vectors with populations accelerate extraction of the top principal components?

**Core ML.** EigenGame could be useful as a diagnostic or for accelerating training (Desjardins et al., 2015; Krummenacher et al., 2016); similarly, spectral normalization has shown to be a valuable tool for stabilizing GAN training (Miyato et al., 2018).

Lastly, GANs (Goodfellow et al., 2014) recently reformulated learning a generative model as a two-player zero-sum game. Here, we show how another fundamental unsupervised learning task can be formulated as a $k$-player game. While two-player, zero-sum games are well understood, research on $k$-player, general-sum games lies at the forefront in machine learning. We hope that marrying a fundamental, well-understood task in PCA with the relatively less understood domain of many player games will help advance techniques on both ends.

## ACKNOWLEDGEMENTS

We are grateful to Trevor Cai for his help scaling the JAX implementation of EigenGame to handle the large IMAGENET experiment and to Daniele Calandriello for sharing his expert knowledge of related work and advice on revising parts of the manuscript.

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
