# OpenReview forum: "EigenGame: PCA as a Nash Equilibrium"
_ICLR.cc/2021/Conference — ICLR 2021 Oral_

### Official Review · AnonReviewer1 · 2020-10-29
**PCA as a Nash equilibrium**

**Rating:** 7
**Confidence:** 3

**Review:**

The authors present new insights on PCA analysis by reconceiving it in terms of a Nash equilibrium among different players, related to the different components. The importance of an objective function minimizing the off-diagonal elements of R is emphasized. The insights lead to parallel algorithms and are demonstrated on large scale problems, which is nice. Overall the new insights can be very valuable and also inspiring for future work and for new developments,  from a broader perspective.

Points that can be improved:

- related to eqs (1)(2) the authors claim that there is a problem with some classical interpretation of PCA analysis. However, this statement is unclear and possibly incorrect: for coming to (2) the authors start from the solution i.e. the eigenvalue problem, while this should be the result of the derivation. This part should be clarified. Probably it is better to replace this part of the paper by a standard formulation and derivation of PCA analysis as given in standard textbooks. Also one can find it component per component and add orthogonality constraints in each step.

- it is nice that a parallel algorithm is obtained. However, at this point the positioning of the result within the existing literature is not clear yet. In the areas of signal processing and neural networks, parallel algorithms and parallel implementations (systolic arrays, VLSI, etc.) of PCA analysis exist for more than 30 years, see e.g.  the book Principal Component Neural Networks: Theory and Applications by K.I. Diamantaras, S.Y. Kung, and related work.

- section 5: missing methods are e.g. the power method, Lanczos method, Jacobi etc.

- related to future work the authors mention the role of PCA analysis with respect to VAE. There is existing work on generative kernel models, eigenvalue problems and disentanglement that is related to it, see Pandey et al  https://arxiv.org/abs/1906.08144

---

> ### Author Response · Authors · 2020-11-13
> **Response to Reviewer 1**
>
> Thank you for pointing out these areas of improvement. We agree, it is difficult to do justice to the wealth of PCA research in the space allotted. We will use our additional space to expand on related work.
>
> We do not claim there is a problem with the classical interpretation of PCA. We claim only that recent work in machine learning focuses on learning the top-k subspace as opposed to the top-k principal components. For example, two popular ML texts, “Understanding Machine Learning: From Theory to Algorithms” by Shalev-Shwartz and Ben-David ‘14 and “Pattern Recognition and Machine Learning” by Bishop ‘06, both present PCA as learning only the top-k subspace. In contrast, a text dedicated to PCA such as “Principal Component Analysis” by Jollife ‘02 introduces PCA as the sequential procedure you sketch above and it is stated that “the standard derivation of PCs, [is] in terms of eigenvectors of a covariance matrix”. In our opinion, presenting PCA as an eigenvalue problem is more clear and concise than presenting it as the endpoint of a chosen algorithm. We absolutely do not claim the eigenvalue problem formulation as a contribution or result of our derivation. Please let us know of any statements we make that give that impression. We will edit the paper to clarify our contribution.
>
> Our paper does not address or optimize hardware design for PCA (systolic arrays, VLSI), which is likely important for squeezing the most out of our proposed parallelizable algorithm. An extensive comparison of distributed/decentralized implementations is beyond the scope of this paper. Our focus is on the core algorithm and we leave specific details of how best to implement it to future work. Note that our algorithm was straightforward to implement and scale to very large problem sizes using readily available hardware and frameworks.
>   That being said, reviewer 4 had a similar concern and asked specifically for a discussion comparing against a distributed GHA algorithm. Please read our response above (see the bullet "Decentralized" in the second part of our response) and let us know if that is the sort of discussion you had in mind. We will add text to acknowledge the extensive prior research that has been done to parallelize algorithms for PCA.
>
> We will expand the related work section to include more approaches. Note however that the Lanczos method does not directly compute eigenvectors. It requires an additional computation step whereby the eigenvectors of a tridiagonal matrix are efficiently computed. The power method computes the top eigenvector and can be extended via deflation to compute all the eigenvectors. We will also include a reference to the Jacobi eigenvalue algorithm.

---

### Official Review · AnonReviewer3 · 2020-10-30
**A novel decentralized PCA derived from the perspective of Nash equilibrium.**

**Rating:** 8
**Confidence:** 3

**Review:**


A. Summarize
This paper proposes to not only maximizes the trace of the projected covariance matrix R but also minimizes the off-diagonal element of R, which helps to recover the real principal components(eigenvectors of the covariance matrix) from data, while the other large-scale algorithms only recover the top-k subspace.
Furthermore, the authors utilize the hierarchical relation between eigenvectors and design a utility function for each eigenvector. Therefore, each eigenvector serves as a player in a game and they will achieve strict-Nash Equilibrium at the end, which enables a decentralized algorithm for large-scale PCA problems.
In the experiment, the authors conduct experiments on synthetic data, moderate scale data, and large scale data by resnet activation maps. The first two experiments demonstrate that the proposed algorithm is competitive with Oja's algorithm and even better under some conditions. The large scale experiment on resnet activation maps is only feasible by the proposed algorithm and demonstrate that it is a powerful tool to achieve interpretable representation.

B. Strength
1.  This paper is well organized and easy to follow. The explanation about why other algorithms only recover the top-k subspace but not the principal components is step-by-step. Based on this observation, the authors propose to minimize off-diagonal elements and derives the utility function, which adds a generalized Gram-Schmidt step to the gradient and can be decentralized naturally.
2. Besides the algorithm itself, the proposed method also opens new doors for other interesting future research other than recovering principle components.
3. The experiments are simple yet sufficient to demonstrate the superiority of the proposed algorithm. To the best of my knowledge, the proposed algorithm is the first that dealS with a problem as large as in the resnet-200 experiment.

C. Weakness:
1. My only question is that the proposed algorithm focuses on recovering the real principle components and finding interpretable features. So it would be good if we can see some comparison of the lower-dimensional features in some downstream applications. For example, can we cluster the input data into meaningful clusters better than other algorithms?

D. Justification of score:
This is a great paper that gives a new perspective on PCA and derives a novel decentralized large-scale algorithm and will inspire a lot of further research along this line. So I vote to accept this paper.

---

> ### Author Response · Authors · 2020-11-13
> **Response to Reviewer 3**
>
> Thank you for your positive review. We are pleased to hear you found the paper was easy to follow, especially Section 2 and that you generally enjoyed the paper.
>
> The feature space recovered for the synthetic experiments are not interesting and PCA has been run on MNIST many times, but to our knowledge, we are the first to compute the PCs of the activations of a ResNet-200. We could also add to the appendix 3D scatter plots of activations projected onto different subsets of the computed PCs if that would help. We are also considering including experiments of a “downstream problem” where we measure the performance of a KNN classifier trained on the feature space defined by the principal components. This is not necessarily a practical approach to classification of Imagenet, but it would help reveal whether or not the PCs of the ResNet-200 activations provide useful features.

---

### Official Review · AnonReviewer4 · 2020-11-02
**A comprehensive and interesting take on the PCA problem; needs some more refinement, which can possibly be managed in a revision**

**Rating:** 8
**Confidence:** 3

**Review:**

Principal component analysis (PCA) is a well-known dimensionality reduction and feature learning technique in the literature that leads to uncorrelated features. While there are a plethora of algorithms for PCA, along with accompanying analysis, a majority of these works have been developed from an optimization perspective. This paper differs from existing works in that it motivates the $k$-PCA problem, which involves learning the $k$-dominant eigen vectors of the sample covariance matrix, as a competitive game between $k$ players in which each player is supposed to estimate one of the eigen vectors and the PCA solution is the unique strict-Nash equilibrium. The main contributions of the paper in this regard are the following:

- Setting up the PCA problem as a competitive game between $k$ players and showing that the Nash equilibrium corresponds to the PCA solution (Theorem 2.1)
- Development of two games (algorithms), with one a sequential algorithm and the other a decentralized algorithm, for solving the PCA problem (Algorithms 1 and 2)
- Convergence analysis of the sequential algorithm under a restrictive set of assumptions (Theorem 4.1)
- Establishment of the equivalence between the decentralized algorithm and the Generalized Hebbian Algorithm (GHA) of Sanger (Proposition H.1)

Overall, this is a novel paper in that it offers an alternate view of the PCA problem, which might lead to further advances in our understanding of PCA-type algorithms in the future. I therefore have a favorable view of this paper. There are however several important aspects of this paper that need to be clarified by the authors in a subsequent revision before it becomes ready for publication.

**Major Comments**

1. Theorem 2.1 is based on the assumption of the top-$k$ eigenvalues being distinct. Algorithms such as Orthogonal Iteration ("subspace" power method), to the best of my understanding, only require an eigen gap between the $k$ and $k+1$ eigenvalues and do not require the first $k$ eigenvalues to be distinct. This needs to be discussed clearly in the paper.
2. While Theorem 4.1 for the sequential game does not explicitly state it, it appears that it also requires the eigenvalues to be distinct (Theorem L.4, e.g.). This, once again, is a major assumption that is neither discussed clearly in the paper, nor compared to other works that do not seem to have this limitation.
3. Majority of the works in the PCA literature require the initialization subspace to not be orthogonal to the $k$-PCA subspace. This work, however, requires the stringent assumption that each eigenvector is initialized to within $\pi/4$ radians of the original eigenvector. Not only is this a strict probabilistic assumption in the case of random initialization, but it also becomes harder to satisfy as $k$ increases (as the authors also discuss). In light of this strict condition, this reviewer is confused by the claim in the paper that "these theoretical findings are strong relative to other claims." I would also have liked the authors to discuss this assumption up front in the paper.
4. The sequential game appears to be very similar to other approaches that have been proposed in the literature that estimate an eigenvector, subtract its contributions from the data, and then estimate the next eigenvector (see, e.g., Allen-Zhu and Li, 2017 and Raja and Bajwa, 2020). Such approaches of course suffer from the fact that they require distinct eigenvalues, but they don't require any QR decomposition. There is however no discussion of the connections between such approaches and the proposed sequential game.
5. Why is the decentralized game being called "decentralized"? Is there a distinction the authors are making between distributed and decentralized? What's the topology being considered by the authors in relation to this game and what exactly does "broadcast($\widehat{v}_i$)" mean in terms of reaching out to other nodes? Some discussion of this would be useful.
6. While the decentralized game has not been analyzed in this paper, the distributed variant of GHA has been analyzed in the literature; see "Fast and communication-efficient distributed PCA". It would be helpful for the authors to comment on the differences between their decentralized algorithm and this distributed GHA work.
7. Why is "the longest streak of consecutive vectors with angular error less than $\pi/8$ radians" the right metric for the experiments?
8. The claim in Figure 3 that "We omit Krasulina’s as it is only designed to find the top-$k$ subspace" is not clear to this reviewer.
9. It would be useful for the authors to discuss the use of $\nabla^R_v$, rather than $\nabla_{v_i}$, for updates in both algorithms.

**Minor Comments**

1. In my opinion, it is incorrect to say that PCA leads to interpretable features.
2. The claim "An exponential convergence rate in the full-batch setting is possible using Riemannian acceleration techniques" is perhaps too ambitious, unless the authors are confident that this is doable, in which case one wonders why this was not shown in the paper.
3. The experimental plots in the paper is too hard to see clearly. It might be a useful idea to add them to the appendix also, where they can be shown in larger sizes.

***Post-discussion period comments***

The authors have satisfactorily addressed all of my comments as well as, in my opinion, comments of other reviewers. Based on the latest revised version of the paper, I am increasing my score to 8 (from 7). I believe this paper is worthy of publication in proceedings of ICLR 2021 and I recommend it as such.

---

> ### Author Response · Authors · 2020-11-13
> **Response to Reviewer 4**
>
> Thank you for taking the time to give our paper such a thorough review. Our manuscript will greatly benefit from your comments. We have grouped your comments by theme below and have answered them to the best of our ability. We would very much appreciate it if you could follow up with us on any remaining open questions or concerns.
>
> First, regarding your summary bullet on GHA vs EigenGame, to be clear, these algorithms are not the same. Proposition H.1 points out a similarity between them, but they have very different performance as can be seen in Figure 3 (blue vs maroon). Please let us know if this is a point of confusion.
>
> Major:
>
> EigenGap (1, 2) -- All results in the paper assume the eigenvalues are distinct. Thank you for raising this issue. It seems we only state this once in the main body (Theorem 2.1). We will add it to the problem formulation in Section 2. The main reason we present EigenGame with this assumption is to ensure there exists a unique ground truth for the eigenvectors of the covariance matrix. If there are repeated eigenvalues, then the PCs are not unique and measuring alignment to them (as we do in experiments via “Longest Streak”) is undefined.
>   This being said, we did not observe repeated eigenvalues among the top-k for MNIST nor for the activations of the ResNet-200. For dense real-valued data, it is highly likely that the eigenvalue gaps are extremely small, but still nonzero.
>   Allen-Zhu and Li (‘17) provide a gap-free convergence guarantee for the individual Rayleigh quotients of the returned (orthogonal) eigenvectors. We believe this type of guarantee is flexible enough to be meaningful for both distinct and repeated eigenvalues. For distinct eigenvalues, this implies convergence to the unique eigenvectors. For repeated eigenvalues, this implies the eigenvector has converged to the correct subspace (which is the best that can be achieved given the eigenvectors in this subspace are not unique). We expect that EigenGame (Algorithm 1) satisfies this guarantee as well because 1) our utilities equal their Rayleigh quotients at convergence and 2) Corollary L.9.1 plus the Riemanian convergence guarantee in Lemma L.1 relate convergence to optimality of the utility function. Note that the Nash equilibrium would no longer be “strict”. We have rerun our synthetic experiments with a “bubble” of repeated eigenvalues in the spectrum (eigenvalues 10-19 are the same) and found that EigenGame accurately returns the unique eigenvectors on either side of this “bubble” (50 eigenvalues in total). Despite our confidence, we consider extending our proof and experiments to handle the case of repeated eigenvalues to be outside the scope of the paper. We believe the paper stands on its own even with this simplifying assumption.
>
> Initialization (3) -- This is a misunderstanding. Convergence of EigenGame does not require ANY assumptions on initialization. We state this in the first sentence of Theorem 4.1. To be concrete, given any initialization, step 2 of Algorithm 1 calculates the maximum number of iterations ($t_i$) required to approximate an eigenvector to a user-given tolerance ($\rho_i$). This number is finite but may be arbitrarily large depending on how close the angle between the initialized vector and the true eigenvector is to 90 degrees. This type of universal convergence guarantee is less common in the literature, which is why we stress our claims are strong.
>   If the vector happens to be initialized within 45 degrees, we provide in the latter half of Theorem 4.1, a finite sample convergence rate that sheds light on how convergence depends on the properties of the spectrum. In contrast, Matrix Krasulina by Tang ‘19 guarantees an exponential convergence rate if the initial vectors capture the top-(k-1) subspace. Note this is a much stronger condition than requiring the “initialization subspace to not be orthogonal to the k-PCA subspace”. This is not necessarily representative of the entire PCA literature. We’re just using this example to better place our contribution.
>
> Sequential Approaches (4) -- We will use the additional space to expand the related work section, specifically to cover sequential and deflation-based approaches. GHA represents an approach that blends the sequential approach analogously to how Algorithm 2 blends the sequential steps of Algorithm 1.
>   Regarding the two references you provide, to our understanding, Allen-Zhu and Li present Oja and Oja++ both relying on a QR step and are not sequential algorithms. Also, Raja and Bajwa only analyze solving for the top eigenvector and so a sequential approach is not considered there. Are those the references you meant to provide? Note, we cite both these papers in the related work section.

---

> > ### Author Response · Authors · 2020-11-13
> > **Response to Reviewer 4 (continued)**
> >
> > Decentralized (5, 6) --  The exact topology we consider is presented in Figure 1. Each eigenvector $v_i$ “lives” on its own node/machine. Not pictured with each node is its own datastream. Also, the “broadcast($v_i$)” step is explained in Figure 1; each node must broadcast its vector $v_i$ to the other nodes down the hierarchy so that they may compute the necessary gradients.
> >   Thank you for pointing out the work on distributing GHA/Sanger’s algorithm. That work replicates the entire PCA computation across several nodes, each of which is observing an independent data stream. In contrast, we consider distributing a single PCA computation across the eigenvectors, so our approach is orthogonal. Replicating EigenGame (with all k vectors) across several nodes and sharing information between the EigenGames would be analogous to the approach explored in the distributed GHA paper. In other words, replace each node in the distributed GHA paper with EigenGame and then expand each EigenGame node to reveal that the game is additionally distributed across k eigenvectors. We referred to this as decentralized as we don’t require a “master” node for coordinating the computation. If after our explanation above, you think this is better described as distributed, please let us know and we will consider changing our terminology.
> >
> > Longest Streak (7, 8) -- The threshold of $\frac{\pi}{8}$ for angular error was chosen somewhat arbitrarily. We can add experiments to the appendix for $\frac{\pi}{16}$ and $\frac{\pi}{32}$. We present the longest streak as opposed to “# of eigenvectors found” because, in practice, no ground truth is available and we believe the user should be able to place higher confidence in the larger eigenvectors being correct. If an algorithm returns 100 vectors, 50 of which are accurate PCs but does not say which, this is less helpful. This is not the only important metric. We also plot subspace distance.
> >   Krasulina's is not designed to return an ordered set of eigenvectors. It is designed to find a set of vectors that spans the top-k subspace, therefore, it expectedly does poorly on the longest streak metric (aka returns a streak of zero). We have results for it and can include them, but they are similar to the MNIST experiment.
> >
> > Riemannian Opt (9) -- Thank you for pointing this out. That discussion appears to have been omitted from the submission. $\nabla^R$ projects the gradient onto the tangent space of the sphere (a Riemannian manifold) - $R$ stands for Riemannian. This gradient projection step is illustrated and discussed in more detail in section G of the appendix.
> >
> > Minor:
> >
> > Interpretability (1) -- We agree that PCA does not, in general, lead to interpretable features (unless possibly if the data is Gaussian). Interpretability is difficult to define, but among all the possible orthogonal bases for the top-k subspace, the one that aligns with the directions of maximum variance is at least chosen according to a principle, and this makes the resulting PCs interpretable according to that metric. In contrast, the individual features given by a randomly chosen basis for the top-k subspace are, by definition, not chosen according to any principle and hence are less interpretable. We can change our statement to indicate the eigenvectors are $more$ interpretable than a random basis for the top-k subspace.
> >   In future work, it would be interesting to consider extending our game-ified approach to a similar problem, non-negative matrix factorization, which has been found, in practice, to extract sparse and easily interpretable factors from data.
> >
> > Exponential Convergence (2) -- We will leave this out. Accelerated Riemannian optimization techniques are sufficiently new that incorporating them would be a nontrivial extension.
> >
> > Plots (3) -- We will add enlarged versions of Figures 3 and 4 to the appendix.

---

> > > ### Comment · AnonReviewer4 · 2020-11-24
> > > **The response and the November 23rd revision are quite useful ... some minor comments**
> > >
> > > I would like to thank the authors for their detailed response. The latest revision (November 23) looks quite good. Here is my feedback on the authors' response to my earlier comments.
> > >
> > > -  **RE:** GHA vs EigenGame. Yes, I agree that the summary bullet was not worded in the best of manners.
> > > - **RE:** EigenGap. I agree that extension to an eigengap free result can be part of future work. Perhaps a brief statement to this effect in the paper would be helpful to other researchers.
> > > - **RE:** Initialization. I appreciate the clarification. Should the authors change their phrasing on page 6 to "... these theoretical findings are strong relative to ***some*** other claims"? Orthogonal iteration (despite its computational challenges), e.g., does not require such an assumption to give us non-asymptotic convergence guarantees.
> > > - **RE:** Sequential schemes. I stand corrected on this point as Allen-Zhu and Li indeed requires QR decomposition and Raja and Bajwa only talks about the k-PCA problem for k > 1 in passing.
> > > - **RE:** Decentralized. Thanks for the clarification. I understand now and decentralized is fine in this case, except that there is a specific topology that is being enforced within this algorithm.
> > > - **RE:** Longest Streak. It is certainly an interesting choice of metric. Has it been used before in the literature? My assumption is that's not the case?
> > > - I am in agreement with the remaining part of the authors' response.
> > >
> > > There is a small typo in the paper on page 2: "We then consider learning only **the the top-$k$** and ..."

---

> > > > ### Author Response · Authors · 2020-11-24
> > > > **Thanks for your keen eye. Minor comments addressed.**
> > > >
> > > > We made a few more changes according to your comments
> > > > - Added "We leave proving convergence in this setting to future work." to the paragraph discussing the "bubble" experiment,
> > > > - Added the qualifier "some" --> "some other claims",
> > > > - Removed the second "the" in "the the top-k".
> > > >
> > > > Regarding longest streak, no, we have not seen this in the literature before. That being said, we also have not seen eigenvector error measured in terms of angular error either, although it has surely been measured by someone. Also, we added a few sentences (similar to our response above) to the first paragraph of the experiment section to clarify this choice of metric:
> > > >
> > > > "We present the longest streak as opposed to “\# of eigenvectors found” because, in practice, no ground truth is available and we think the user should be able to place higher confidence in the larger eigenvectors being correct. If an algorithm returns $k$ vectors, $\frac{k}{2}$ of which are accurate components but does not indicate which, this is less helpful."

---

### Author Response · Authors · 2020-11-23
**Changes to the Paper**

We have made a few changes to the paper in line with some of our responses below. Specifically, we
- Fixed the typos noticed by the reviewers and made a few minor edits to the writing,
- Added an explanation of $\nabla^R$ and Riemannian gradients to Section 3, in the paragraph adjacent to Figure 2,
- Repeated Figure 3a with an angular tolerance of $\frac{\pi}{32}$ instead of the original $\frac{\pi}{8}$,
- Repeated Figure 3a with 10 repeated eigenvalues in the middle of the spectrum to show EigenGame still converges to the correct eigenvectors on either side of the spectrum. Any orthogonal basis spanning the subspace corresponding to the 10 repeated eigenvalues is a PCA solution, so we do not measure eigenvalue error among those vectors,
- Added Matrix Krasulina to the synthetic experiments to confirm that the results are similar to the MNIST experiment (and uninteresting),
- We expand the related work section (Section 5) to include a discussion of deflation, the Power method, the Jacobi algorithm, and others.

---

### Decision · Program_Chairs · 2021-01-07
**Final Decision**

**Decision:**

Accept (Oral)

**Comment:**

This paper introduces a novel game-theoretic view on PCA which yields an algorithm (EigenGame; Algorithm 2) that allows evaluation of singular vectors in a decentralized manner. The proposed algorithm is significant in its scalability, as demonstrated in the experiment on a large-scale dataset (ResNet-200 activations). This paper is generally clearly written, and in particular Section 2 provides an easy-to-follow reasoning leading to the proposed game-theoretic reformulation of PCA. I felt that the later sections are a bit condensed, including the figures. In the authors response major concerns raised by the reviewers have been appropriately addressed. I would thus recommend acceptance of this paper.

What I found particularly interesting in their game-theoretic reformulation is that in the utility functions shown in (6) the orthogonality constraints $\hat{u}_j^\top\hat{u}_i=0$ have been removed and replaced with the soft constraints represented as the regularizer terms encouraging the orthogonality. Although several alternative forms for the regularizers would be possible, it is this particular form that allows an efficient gradient-ascent algorithm which does not require explicit orthonormalization or matrix inversion is straightforwardly parallelizable.

Pros:
- Provides a novel game-theoretic reformulation of PCA.
- Proposes a sequential algorithm and a decentralized algorithm for PCA on the basis of the game-theoretic reformulation.
- Provides theoretical guarantee for the global convergence of the sequential algorithm.
- Demonstrates that the proposed decentralized algorithm is scalable to large-scale problems.

Cons:
- The latter statement of Theorem 4.1 requires conditions on the initialization, which are hard to satisfy in high-dimensional settings.
- Significance of the proposed game-theoretic formulation in the context of game theory does not seem to be well explored.